# Managing incontinence in low-and middle income-countries: A qualitative case study from Pakistan

**Zara Ansari[1], Sian White**[2]*

**1** Currently with Acasus, Karachi, Pakistan, **2** Department of Disease Control, London School of Hygiene and Tropical Medicine, London, United Kingdom

* Sian.white@lshtm.ac.uk

## Abstract

### Background

Incontinence is a complex health and social issue, which involves the involuntary loss of urine or faeces or both. Individuals with disabilities are particularly vulnerable to incontinence. The management of incontinence has largely been overlooked in low and middle-income settings (LMICs). This study aimed to explore the incontinence management strategies employed by disabled people with severe incontinence and their caregivers in Sindh Province, Pakistan.

### Methods

Incontinence management was explored through in-depth interviews with people with incontinence (PWI) and their caregivers, photovoice, and a market survey and product attribute assessment. Data was analysed thematically through inductive coding and evaluated against existing disability and caregiver frameworks.

### Results

Incontinence management affected all aspects of daily life for PWI and caregivers. Effective management of incontinence was prioritised because caregivers viewed it to be part of their familial duty and a requirement for the household to remain pure in the eyes of God. Coping strategies included strict adherence to routines, reducing food and drink intake, creative uses of locally available natural resources, and a heavy reliance on soap and water for maintaining hygiene. Products such as adult diapers were largely unavailable, costly and were not deemed suitable for regular use. There were no social or medical interventions in the region to support incontinence management. The main impacts of incontinence on the household were social isolation, stigma, role shifts within the family, the development of physical ailments among caregivers, and decreased income.

### Conclusion

The complex health, psychological, social, economic, and cultural impacts of incontinence are exacerbated in LMICs due a lack of recognition of the condition, the absence of social or

**Data Availability Statement:** Interview transcripts cannot be shared publicly because of confidentiality and risks of deductive disclosure. Data is available upon request from the authors (contact SW) or from the Reserach Governance

and Intectity Office at LSHTM: rgio@lshtm.ac.uk. The data is available for researchers who meet the criteria for access to confidential data.

**Funding:** The authors received no specific grant funding for this work. Travel and publication costs were supported by Norwegian Church Aid and the London School of Hygiene and Tropical Medicine. The funders had no role in study design, data collection and analysis, decision to publish, or preparation of the manuscript.

**Competing interests:** The authors have declared that no competing interests exist.

medical interventions and limited access to basic WASH infrastructure, and assistive devices or products. Appropriate solutions need to be developed in partnership with PWI and caregivers and need to be contextualised, affordable and sustainable.

## Introduction

Urinary and faecal incontinence is a complex public health issue with a range of social and health consequences. Urinary incontinence (UI) is defined as any involuntary loss of urine [1] whereas faecal incontinence (FI) is the inability to control bowel movements, resulting in the involuntary passage or leakage of stools [2]. Incontinence most commonly effects older people, multiparous women, children and people with physical or cognitive impairments but can impact anyone at any life stage [3, 4]. The global burden of incontinence is difficult to estimate. Most epidemiological studies are from high income countries [5–7], where the prevalence of UI is estimated to be 28% in women, and 10% in men.

Incontinence impacts the physical and mental health status of the individual. Incontinence is associated with urinary tract infections, higher exposure to feacal-oral pathogens, increased number of bacterial and fungal skin infections and higher prevalence of depression and anxiety [5, 8, 9]. The health-related quality of life (HRQoL) impact of incontinence is similar to that observed with other chronic medical conditions like osteoarthritis, chronic obstructive pulmonary disease, and stroke [5].

The shame and public stigma of incontinence affects treatment seeking behaviour [10–14] with most people who experience incontinence not seeking treatment because of embarrassment or low expectations of treatment outcomes. Incontinence can impact a person's ability to fully participate in society, for example in the workplace it can reduce working hours and have negative impacts on concentration and performance of physical activities [15].

Research from high income countries has shown that incontinence management places significant economic burdens on both the patient and the health system [14]. In 2000, the total cost of managing incontinence in the US was estimated to be USD19.5 billion [16].

In high income countries, a wide variety of products and support services are available to manage incontinence. These products are available in pharmacies and supermarkets or even online. These include but are not limited to disposable diapers or inserts, reusable pull up pants, disposable catheters and disposable mattress protectors [14]. In high income settings water and soap are typically readily available—both of which are key in effectively managing incontinence.

While little is known about how incontinence is managed in low-and middle-income countries (LMICs), a recent study in Vanuatu highlighted a range of barriers including lack of incontinence products or assistive devices, social stigma towards the condition and isolation of people with incontinence [17]. This study was designed to understand the coping mechanisms employed by older people or people with disabilities and their families to manage incontinence in a LMIC setting. The study also explored the relative importance incontinence management in comparison to other issues of daily life for older people or people with disabilities.

## Methods

### Study site

This study was conducted in urban and rural areas of Sindh Province in Pakistan in July 2017. According to the 2004 World Health Survey, the prevalence of disability in Pakistan is

approximately 13% [18]. Water, sanitation and hygiene (WASH) coverage varies substantially between urban and rural regions of Pakistan. In Sindh Province 23% of the population has access to safely managed water. In urban areas of the province the majority of people (89%) have access to a basic handwashing facility with soap and water, however in rural areas this drops to 63% [19]. Similarly basic sanitation coverage is estimated to be 82% in urban areas compared to only 33% in rural parts of the province. The data collection for this research occurred during the rainy season when water is typically less scarce. During the period of the study there were population within Sindh who experienced displacement due to flooding. In the rural areas the houses we visited were made from mud and had thatched roofs. The main occupation of people in these rural areas is farming, and so all houses had some form of live-stock (e.g. buffalos, cows or goats) present. In urban areas people typically lived in cement or brick houses and were engaged in a more diverse set of occupations including working in factories, offices or shops.

## Data collection

The study employed a range of participatory qualitative methods including in-depth interviews with people with incontinence and their caregivers, photovoice, a market survey of incontinence products, and field note taking. These methods are described in detail in Table 1.

## Sample and recruitment

Any person with either faecal and/or urinary incontinence was eligible to be included in the study as a primary participant. Hereafter primary participants are referred to as participants with incontinence (PWI). PWI were excluded if their incontinence could be repaired through medical intervention such as women with fistulas or men with prostate issues.

Caregivers of primary participants were also included as 'Secondary participants' if they were actively involved in assisting to manage the PWI incontinence.

**Table 1. Detailed description of participatory qualitative methods.**

| Description | Sample Size |
|---|---|
| **In-depth Interviews:** We developed an open-ended question guide that focused on the management of incontinence. This included questions on incontinence management, coping mechanisms (individual and of their carers), barriers, product use and access to water, sanitation and hygiene (WASH). | 10 people with incontinence and their caregivers (5 of these participant-carer pairs lived in urban settings and 5 in rural settings). In each pair the person with incontinence was interviewed separately to the carer so that they both were able to speak freely about their experiences. |
| **Photovoice:** This aimed to understand the impact of incontinence within the larger context of issues faced by people with disabilities. Respondents were given a camera and asked to take a total of 10 pictures, 5 representing the best part of their day and 5 representing the challenges they face in managing their incontinence. This was followed by asking the participants to rank the pictures from most important to least important. This followed a protocol piloted in previous work with people with disabilities [20]. | 4 people with disabilities (2 urban and 2 rural) |
| **Examination of physical artefacts:** A market survey was conducted to understand the availability of products used for incontinence management. We approached a range of local shops including local pharmacies, general stores and local *thelas* (pushcart vendors) and asked them whether they stocked products for incontinence management. During in-depth interviews, families were asked to bring out all items they were using in the management of incontinence. They were asked a series of questions about each item including how long it has been used for and it's positive and negative attributes. Products purchased in the local market were introduced and a discussion took place on their opinion regarding the products and their perceived cost and utility. | Conducted with the same 10 participant pairs that took part in in-depth interviews. |
| **Field Notes:** The lead researcher (ZA) took extensive field notes during her research. The notes were used to capture key observations about behaviors and coping mechanisms that did not come across in the other methods. | Notes written at the end of each day of data collection. |

In the first phase of sampling, a local disabled person's organization (DPO), health workers, women's groups and village leaders were provided with a definition of incontinence and asked to list any households where they knew of someone that had this challenge. Since there is no direct translation of incontinence in Urdu, or any of the other main regional languages of Pakistan, we developed the definition of incontinence by consulting medical professionals and asking them for a colloquial or medical definitions that they had used when discussing incontinence with their patients. *Peshaab aur paikhane par kaboo na pana* (not being able to control ones urine or faeces) was the closest phrase that was used by most doctors and which was easily understood. However due to a lack of discussion about incontinence, this sampling strategy was amended to also list individuals with cognitive and physical impairments. Families with a person meeting this revised criterion were then visited to see if they were eligible. People with more severe incontinence were intentionally selected to participate, however participants were also purposively sampled from the list to reflect a variation in age, urban-rural location, socio-economic status and impairment type.

## Data management and analysis

Information gathered from all methods was audio-recorded and then transcribed orthographically in Urdu, reproducing all spoken words and sounds including hesitations, cut-offs in speech, laughter and long pauses. All transcripts were imported into NVivo 10 and thematically analysed using a six-step analysis method [21] process which included data familiarisation followed by the development of a coding tree. In our case we inductively developed the coding frame based on the phrases expressed by the participants [22]. The codes were reviewed and constant comparisons were done to develop emerging themes. Themes were summarised and illustrated through quotes. As we wrote up the findings we visualised insights by comparing them to existing disability and caregiver frameworks [23, 24].

## Ethical approval and consent

The study received ethical approval from the London School of Hygiene and Tropical Medicine (ID:13595) and the Interactive Research and Development's Independent Institutional Review Board in Pakistan (ID: IRD_IRB_2017-05-002). All participants and caregivers provided informed written consent. A particularly detailed process of informed consent was done with photo voice participants. This involved PWI being able to select how their images could be used and shared and whether they wished their name or a pseudonym to be used.

## Results

### Characteristics of sample

Table 2 describes the characteristics of the participants and caregivers included in this study. All of the PWI had a physical or cognitive impairment, with majority being male as we could not identify any other primary female participants. All participants were considered to have severe incontinence, and were either unable to use a toilet independently or were reliant on assistive products (e.g. diapers). Of the 10 caregivers, nine were women and all were family members, the majority being wives or mothers of the disabled person. Seven other elderly PWI (without visible physical/cognitive disability) were asked to be part of the study, but all refused to participate. All of these individuals stated that they did not have incontinence, even after their caregivers confirmed otherwise. Photovoice was conducted with four additional participants, all with a physical disability.

**Table 2. Description of interview participants according to their role as either a person with incontinence or a caregivers.**

| | Participants with Incontinence | Caregivers |
|---|---|---|
| Number of participants | 10 | 10 |
| **Age** | | |
| Age range, years | 22–70 | 19–75 |
| **Gender** | | |
| Male | 9 | 1 |
| Female | 1 | 9 |
| **Geographic location of participants** | | |
| Urban | 5 | 5 |
| Rural | 5 | 5 |
| **Nature of Disability** [§] | | |
| **Physical Impairment** | | |
| Paralysis | 7 | |
| Limb Impairment | 4 | |
| Cerebral Palsy | 2 | |
| Polio | 1 | |
| Stroke | 1 | |
| **Cognitive/Intellectual Impairments** | | |
| Intellectual Impairment | 3 | |

[§]Some participants had more than one impairment

## Prioritization of incontinence management within daily life

PWI and caregivers reported that incontinence management was always a priority in their day to day lives. Three themes emerged through interviews and photovoice to explain this prioritization: the importance of routine, the carer's sense of responsibility and the need to be clean in the eyes of God.

**"*Roz mara ka mamol*"—Routine.** PWI and caregivers emphasized that incontinence management affects many aspects of their daily routines. Following a structured daily routine made managing incontinence easier for both carers and PWI. During a typical day caregivers explained that they would normally get up before the PWI so that they could address their own needs. Once the PWI was awake the carer's day was dominated by tasks related to incontinence management. This included bathing the PWI, cleaning their clothes, and washing sheets or bedding from the night before. Over the years, most careers had noticed patterns in the timing of the bowel and urinary movements of the PWI and adjusted routines to align with this:

> "*She usually has to go around 11 and has a schedule. She will then have to go again around 1 30 and will tell me to quickly come home from work to take her because she can't control it otherwise*"

–Caregiver of a PWI, urban area.

Accordingly, most caregivers were also responsible for providing food and drink to the PWI at certain times of the day so that incontinence management could be more easily planned around. Most caregivers were able to describe their daily routine schedules in detail. This level of structure was also described as necessary because activities such as bathing often required the involvement of more than one person.

*"Ye meri zimidari hai"*—**It is my responsibility.**   Within the Pakistani context, providing care to family members was recognized to be part of cultural and religious norms. All caregivers framed their incontinence management work within the discourse of familial duty and responsibility. Specifically, caregivers believed it was their responsibility to preserve the dignity of the PWI they were taking care of. Effective management of incontinence was key to maintaining that dignity, as sitting in urine or faeces was considered humiliating. Even if the task caused embarrassment for the caregiver, their sense of responsibility trumped those feelings:

"*Sometimes I feel ashamed /embarrassed but after all he is part of my family and it is my responsibility to take care of him.*"

–Caregiver of a PWI, rural area.

Even though familial duty drove caregiving behaviour, incontinence management was not always done willingly. One caregiver said she was praised for taking care of her mother-in-law by others in the community, but it was a responsibility she did not want and hoped that her mother-in-law would soon pass away, so that this responsibility would be lifted from her.

*"Paak nahi hain"*—**Not clean for god.**   The interaction between incontinence and faith was another reason incontinence management was prioritized. Religion was seen as a way of making sense of incontinence and disability. Islam is the dominant religion in Pakistan and governs many aspects of behaviour and daily life. Most PWI mentioned that Allah gave them this condition and they felt that they owed it to God to remain clean:

"*We have to have faith in God. Otherwise, what can we do? Whatever God does, that is what will happen. We can't change that. We can be clean to pray to him.*"

–PWI, urban setting

The two Hindu families included in this study also felt that they needed to remain clean in order to practice their religion and emphasized that it was therefore essential to prioritize cleaning and incontinence management over other aspects of their daily life.

## Social, health and economic impact of incontinence

Six themes emerged to explain the impact of incontinence on the PWI and their caregiver.

*"Tanhayee"*—**Isolation and stigma.**   The social impact of incontinence manifested itself as isolation and stigma. While the majority of PWI stated that they did not feel isolated from their family members, a few mentioned relationship changes. The participant below describes how his relationship with his son changed over time:

"*I feel that they have backed off. They don't ask me if they can help me. . ., they keep drifting away from us [PWI and caregiver]. . . and never turn around and ask about their father*"

- PWI, rural setting

The majority of PWI did report feeling isolated from their friends and community members, which resulted in loneliness. PWI and caregivers rarely left their homes and did not participate in most social or cultural events. PWI attributed their isolation to a lack of awareness about their condition and the smell associated with incontinence. This finding also emerged from the photovoice activity with one participant taking image showing him sitting apart from other men in the village. He gave this photo the following caption:

"*They feel like I am a dirty person and do not want to be near me. I can't sit with my friends, most of them have left me and made me feel even more lonely and sad*"

- PWI, rural setting

**"Hamesha Nigrani Karna Parta Hai"—Constant watchfulness.**   A noteworthy social impact for caregivers was social isolation which was experienced and mentioned by all. Some caregivers would only leave the house for a maximum of two hours, provided they could find an alternative person to cater for the participant's needs during the time they were away.

"*The wife is with me all day. She is sitting on the bed with me all day. Whenever I need anything, she is with me all the time. She doesn't sleep. She sleeps for 5 minutes, then I call her and she is with me again.*"

PWI, rural setting

Caregivers were concerned that any lapse in 'watching over' the participant could result in additional work (e.g. the need for additional bathing), negative health consequences (e.g. damage to skin as a result of being stationary and sitting in urinary or faecal matter) and lead to the PWI experiencing feelings of guilt or embarrassment because they were not able to manage their incontinence independently. As such the social impact, isolation and stigma associated with incontinence permeated beyond the PWI to affect the caregiver.

**"Zimidari badalna"- role change.**   Many participants reported that with the onset of incontinence, they experienced a shift in gender-related roles and responsibilities, and that power dynamics within familial relationships changed. Pakistani culture is a patriarchal system which places most social liberties and decision-making in the hands of men, however many of our PWI were male and received care from a female family member. In such circumstances PWI and caregivers described a 'role reversal' that substantially affected their social lives, sense of identity and ran counter to cultural norms. For example, the incontinence management provided by female caregivers led to altered power structures, with female caregivers now making most household-level decisions while male PWI felt completely dependent on them. This role change led to embarrassment in both parties, primarily when it came to bathing. Most caregivers felt that they should never have to see their male family members in such vulnerable positions:

"*I have to hold him like a child when he goes to the bathroom or showers. He is my husband and I should never have to have become this person in his life. No one taught me how to make these decisions.*"

–Caregiver of a PWI, rural setting

"*I am her husband. I was the man of the house and now look at me. I am dependent on her to lift me and clean me. This is not how a man should be. There is no dignity in this. A wife is not meant to do this. I am always ashamed and do not want to bathe.*"

–PWI, rural setting

**"Khana Rokna"—Reducing food intake.**   Reducing food and drink intake emerged as one of the most coping mechanisms to manage incontinence in this setting. This had obvious negative health consequences. For example, it was observed that the majority of PWI were thin and carers explained that they did not want to eat. In Interviews and through photovoice PWI explained that limiting food and drink intake was a way for them to prevent embarrassment

and reduce the burden on their caregivers. One male photovoice participant took a photo of him pushing food away and provided the following caption:

> "*I do stop eating food, especially when there are guests in the house. When the guests come, I stop eating and drinking all water so that I don't go to the bathroom or get dirty in front of them. I have to do my best to not be dirty and smell then*"

> –PWI, rural setting

However, choices about consumption did vary across PWI. Two participants said they increased their water intake to 'balance out' their choice to eat very little, explaining that this was necessary to prevent dehydration during the hottest months of the year (temperatures sometimes got up to 52˚C during the field work).

**"*Peeth main dard hogaya hai*"—Development of physical ailments.** Caregivers frequently reported that they had developed physical ailments because of their caregiving role. The most reported ailment was back and shoulder pains, associated with the regular need to lift and move PWI to bathe and clean them. One male photovoice participant chose to depict the effects caregiving had on his aging mother. He provided the following caption for this image:

> "*She has been taking care of me for so long now, but she can't see properly and walks with her head near the ground. Her back is getting curved taking care of me.*"

> –PWI, rural setting

Caregivers of PWI with cognitive impairments, reported frequent physical ailments and injuries. Maintenance of incontinence was even more difficult because, as one care giver explained, they "*can't carry him like a normal child*". Caregivers for PWI with cognitive impairments also mentioned being injured by impulsive behaviour, primarily when trying to clean the PWI. Sleep deprivation and fatigue were common among caregivers. One of the male PWI's mentioned how his caregivers do not seek healthcare for their health concerns:

> "*Their body begins to hurt, and I do not know how to make it not hurt. I tell them to go to the doctor, but they refuse. I think that I am light, but you don't realize that the body is very heavy and it will have an impact on the people that take care of you.*"

> –PWI, urban setting.

Most PWI had not visited a doctor and those who had did so because of their disability and did not ask about their incontinence, due to embarrassment about the condition. One participant that did speak to the doctor about incontinence, was not given any advice on effectively managing it, being told to simply "*learn to live with it*". Another participant who had spoken to a doctor about his incontinence was provided with a catheter, but he was not told how to use this or any of the health issues that may be associated with its use. PWI explained that they didn't know how to raise the issue of incontinence with health professionals as there is no word for incontinence in the Urdu language.

**"*Paise hi nahi hain*"—No more money.** Financial difficulties came up as a common theme from both PWI and their caregivers. Participants explained that many people in their region lived in poverty—"*we are poor people and have no more money.*" However, they added that their disability and incontinence exacerbated their poverty. Several PWI, especially males, said that they were unable to work or lost their jobs because of their disability and

**Table 3. Incontinence management products and their price in GBP in 2017.**

| Product | Location | Cost† |
|---|---|---|
| **Adult Diapers (Pack of 10)** | Karachi, Hyderabad and Mirpur Khas | £ 5–10 |
| **Catheter** | Karachi and Hyderabad | £20–30 |
| **Plastic Bed Sheets** | Karachi and Hyderabad | £4–5 |
| **Dignity Sheets** | Karachi | £5–8 |

†Conversion rate: GBP £1 = Pak Rs.140

consequently, their incontinence. Caregivers were also faced with a choice between working or full-time care. Some caregivers spoke about selling their livestock and wedding jewellery to gather enough resources to support incontinence care.

## Products for managing incontinence

**Market survey.** Table 3 provides a summary of the products identified through the market survey and their costs. The most commonly available item for managing incontinence was adult diapers, with one brand of diapers being more common as it was made locally. All brands of diapers cost over GBP 5 for a pack of 10 diapers. Most items were available only in urban areas. Disposable pads or inserts, disposable mattress protectors and reusable pull up pants were not available in urban or rural areas.

Shop owners and pharmacists who were asked about products for managing incontinence were often confused by the question. After describing the condition, some knew of adult diapers, but information on products was limited. One pharmacist handed the researcher a box of condoms thinking that this may help with the condition. Others seemed uncomfortable with the request and declined to provide suggestions of relevant products.

***"Pampers se sub kaam nahi chalta"*—Diapers are not the complete solution.** When the products identified through the market survey were shown PWI and caregivers, most mentioned having seen diapers before, even in the rural areas of Sindh. However, when asked about the products they had available for incontinence management none of the five rural households and only two of the five urban households had diapers available and all reported that their use was minimal. Participants explained that diapers could temporarily reduce stress related to incontinence:

"*They are okay. I wouldn't have to worry for a little while. . .. I wouldn't have to be worried about consistently dirtying my clothes then.*"

–PWI, Urban setting.

The problems associated with diapers were seen to outweigh the benefits. All families mentioned that the materials used in diapers meant that they were uncomfortable to use in hot weather with PWI being likely to get heat rashes and blisters from wearing the diapers. Caregivers for individuals with cognitive impairments used diapers more frequently. However, caregivers felt that diapers were not suitably created for adults with disabilities:

"*Even then, it was very difficult for her to keep a diaper on. She didn't want it, said it was uncomfortable. It never fits her properly and then it leaks, so what is the point of having it on if it does not help at all?*"

–Caregiver of an PWI, urban setting.

The cost of diapers also deterred participants from using the product:

"*Each diaper was for Rs 100. If I had to urinate 5 times a day, I will need Rs 500 every day. I didn't ask for it again.*"

PWI, rural setting.

The possibility of creating diapers at home seemed more appealing to participants. Cloth diapers were seen as more comfortable and cheaper alternative for the participants.

***"Saabun aur paani hi chahiye"*—We only need soap and water.** Soap and water were repeatedly identified by PWI and caregivers as the most important products for managing incontinence and associated smells. The majority of the PWI took three to four showers a day to maintain their hygiene and minimize smell, with caregivers estimating that they often had to collect four or five times as much water to support their needs as compared to neighbouring households. PWI who had ample water available reported being happier as their laundry and bathing needs could be met in a timely and convenient manner. In rural areas, physical barriers, such as the distance to water points, were cited as most problematic by some caregivers:

"*It takes a lot of effort to collect water. I have to go collect it almost three times a day and I can never carry enough for my brother's baths. He never feels clean enough and then sits and cries about his smell.*"

- Caregiver of a PWI, urban setting.

In urban areas, some caregivers spoke about the costs of procuring enough water, indicating that this became a substantial part of their household income. All PWI and caregivers explained that soap was key for smell management and general cleanliness. The importance of soap was portrayed through one of photovoice images (see Fig 1).

The majority of PWI and caregivers reported always having access to soap at home. While frequent soap purchasing was recognized as a significant expense, it was worth the cost and some families took loans from local stores to be able to have soap available. Scented soap was considered preferable by PWI and caregivers but was often more expensive.

***"Humari zamin ka samaan"*—Use of local products for incontinence management.** While urban populations relied heavily on soap and water to manage incontinence, rural participants also described the use of straw and grass as a tools for waste management:

"*We have it lying under the bed. When he urinates or defecates, the liquid and some material seeps down into the grass. It is easy to clean as we then use a rake and push it to the side into this pile. The pile is then used to fertilize our crops.*"

- PWI, rural setting.

Rural participants reported using sand as an alternative to soap to clean a PWI hands and body prior to being rinsed with water. Sand or clay were also used to cover the fecal matter so that it dried out and became easier to dispose of. These methods were seen as cost-effective solutions to incontinence management in rural settings. However, caregivers did emphasize that they were only used when they could no longer afford to purchase soap and that sand did not cover up the smell on the participants.

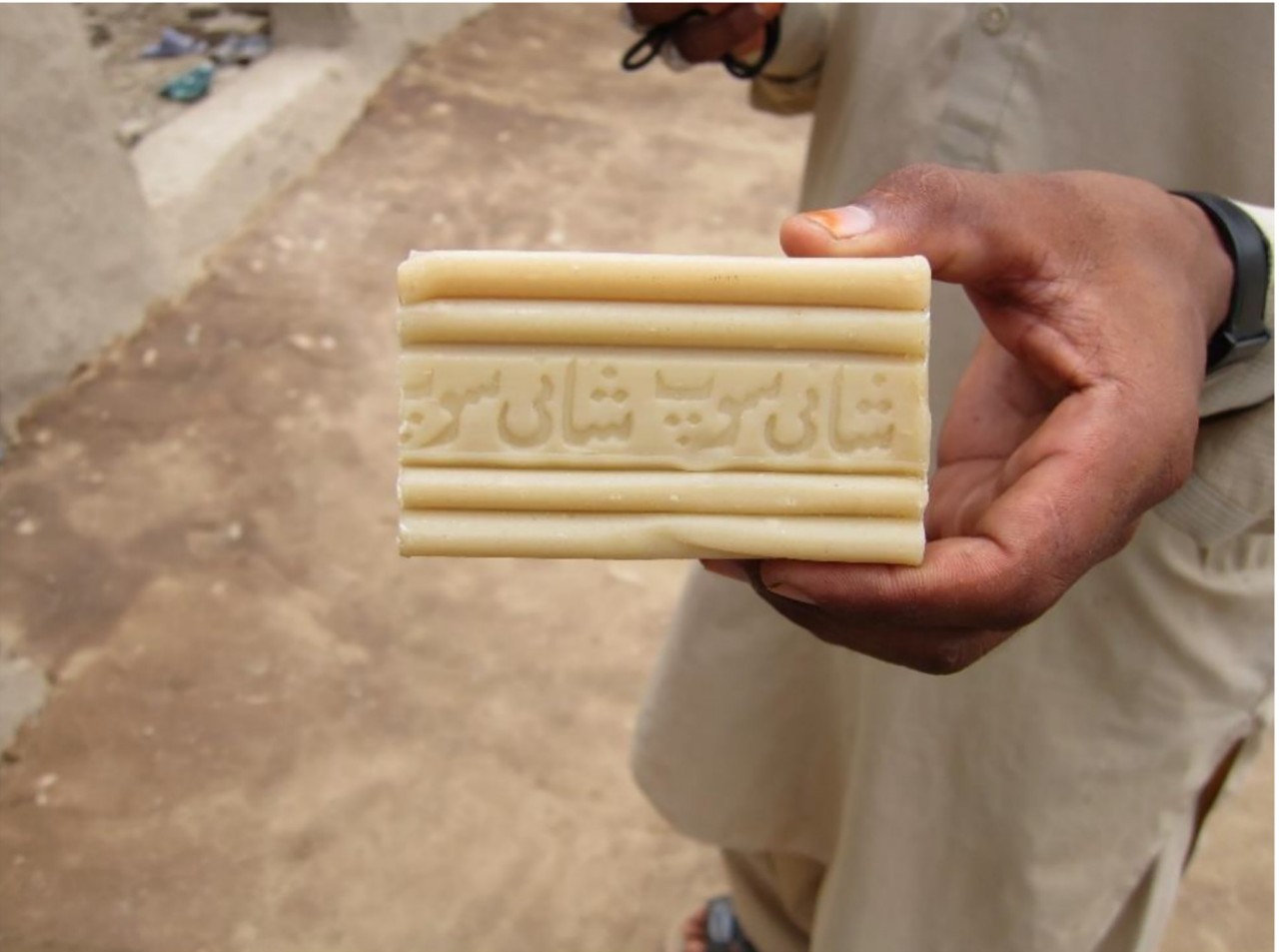

**Fig 1. "This makes me feel very happy and clean.** I use it all the time to manage my smell. It cleans my clothes, and it cleans my hands and the most important thing is that it keeps me clean."—Photo and caption by a PWI, urban area.

## Discussion and recommendations

The emergent themes of this study highlight the need for a holistic approach to the management of incontinence in these settings. Across rural and urban areas of Sindh Province, incontinence was a condition that was largely undiagnosed and rarely discussed. The management of severe incontinence was found to affect all aspects of the daily lives of PWI and their caregivers. In a LMIC setting such as Pakistan, incontinence management created economic hardships for affected households, health challenges, social isolation from the wider community and had negative effects on the dignity, self-esteem and identity of those affected.

Fig 2 summarises the experiences of PWI within our study according to the International Classification of Functioning, Disability and Health (ICF) framework [24]. Fig 3 summarises the experiences of caregivers for PWI according to the Organizing Framework for Caregiver Interventions [23]. Both diagrams illustrate that present challenges with managing incontinence can best be addressed through targeted interventions that make it easier for PWI and caregivers to mitigate socio-economic and environmental barriers.

Our findings are consistent with research documenting experiences of incontinence in high income settings [25–27]. However, challenges related to managing incontinence appear to be

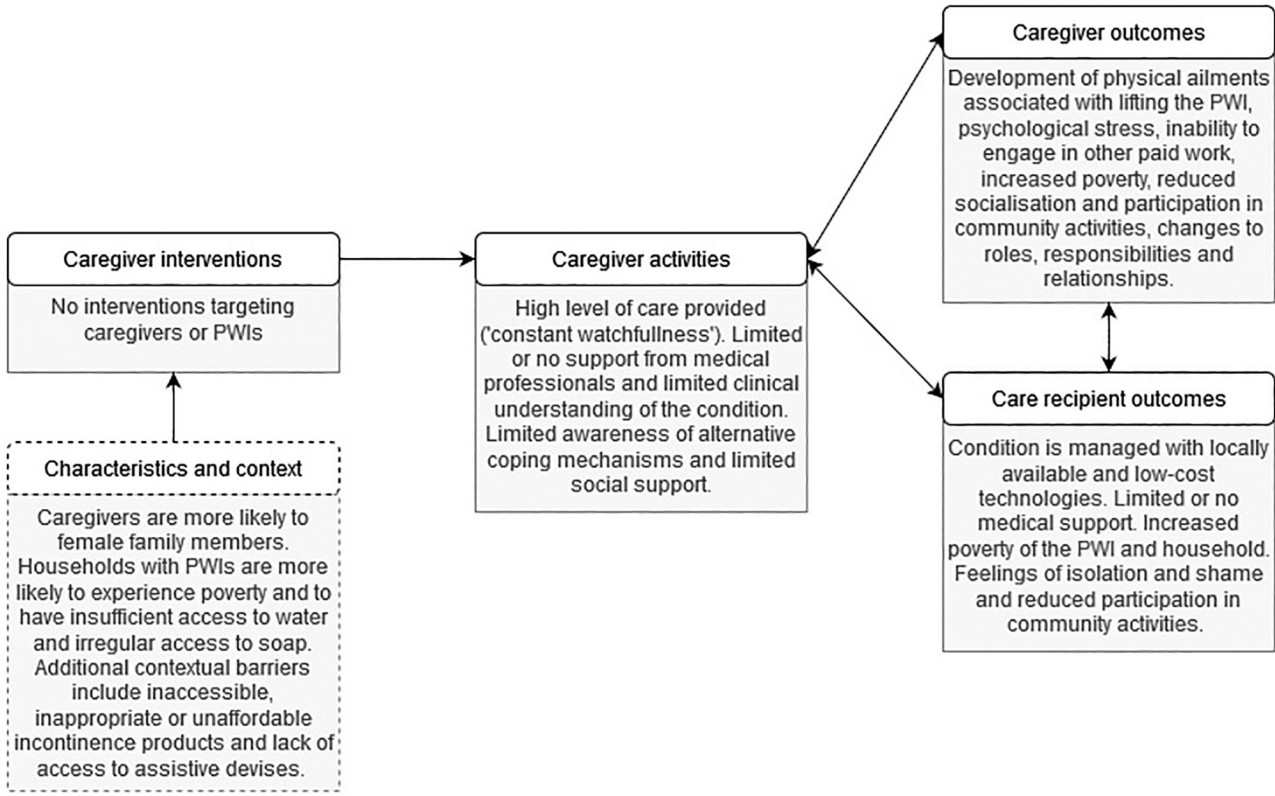

**Fig 2. The International Classification of Functioning, disability and health (ICF) [24] framework adapted to summarise the experiences of people with severe incontinence in LMICs, based on our research.**

exacerbated in LMICs due to factors in the socio-economic and physical environment [17, 20, 28]. For example, in many high-income county settings there are professional care services, assistive products and devices, social protection schemes, health systems and formal or community-based support programmes which can help mitigate the costs and physical and mental toll of caring for a person with severe incontinence [25, 29–31]. While access and use of these services may still vary in high income settings, none of these support mechanisms were available to PWI or caregivers in our study. As such the financial costs and full caring burden were borne by families and tended to compound existing levels of poverty, making such families particularly vulnerable.

Our findings are also consistent with the limited research that exists on incontinence in other LMICs. For example, research in Vanuatu, also found that there was no term for incontinence in the local language and that the condition was rarely discussed with medical professionals [17]. They identified that incontinence products or assistive devises were unavailable or too expensive for regular use, that caregivers often developed physical ailments through caregiving work, and that PWI often reduced food and liquid intake as a key mitigation strategy. The research in Vanuatu suggested that a key mitigation strategy would be to improve access to conveniently located latrines, soap and water for families with a PWI and in public spaces. While this is an important step, our research indicates that for some people with severe incontinence we need to look at solutions beyond latrines. Recognition that toilet use may not be achievable for all is an important consideration given the framing of Sustainable Development Goal 6 [32] which aspires to provide sanitation for all. For our study participants, who

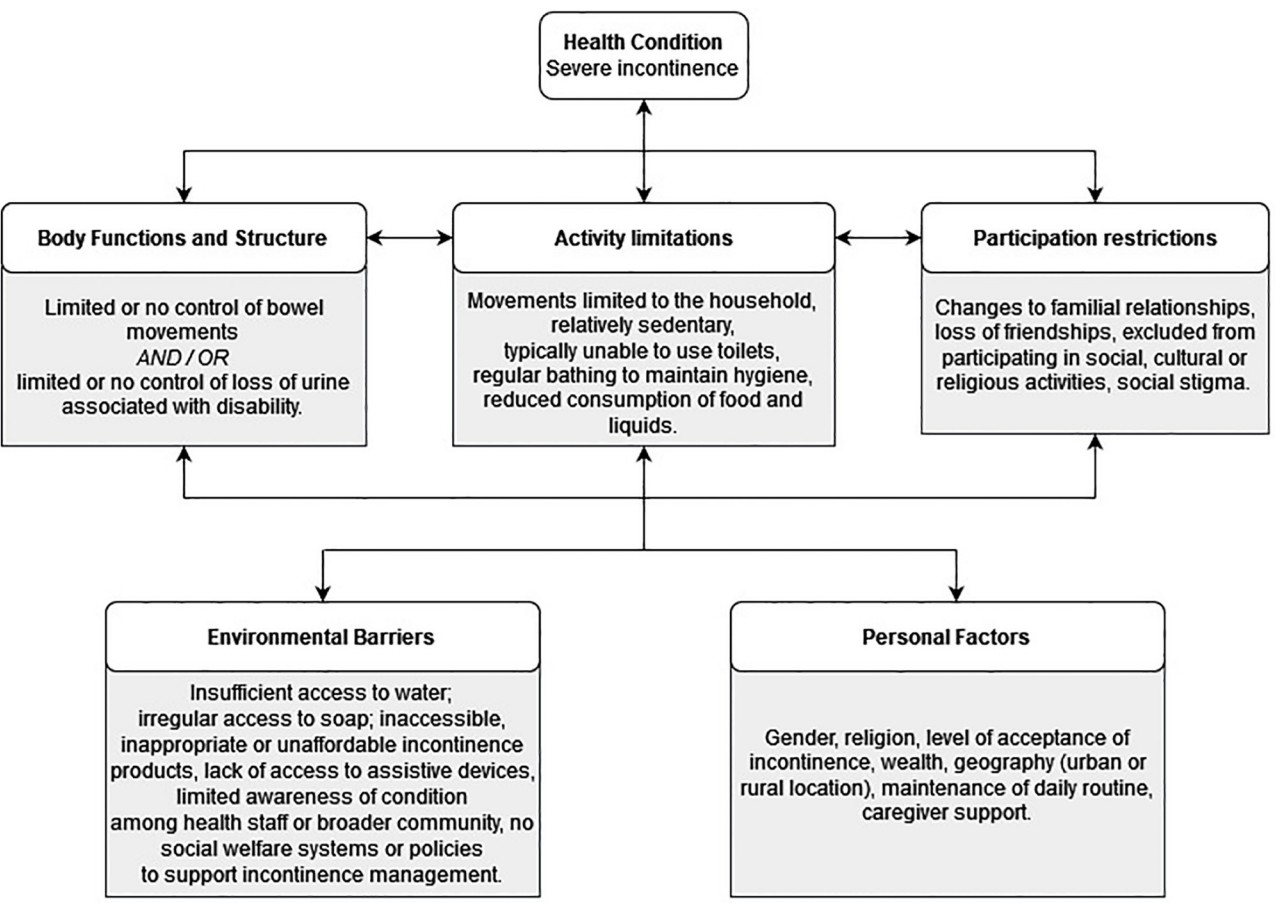

**Fig 3. The organising framework for caregiver interventions [23] adapted to summarise the experiences of caregivers for people with severe incontinence in LMICs, based on our research.**

had limited or no control over their bowel movements or loss of urine, regular bathing with soap and water and other mitigation techniques were more relevant. The increased water requirements needed by our study participants was consistent with findings from a study in Malawi which also found that households with a PWI needed five times as much water [20].

Table 4 describes five domains of potential interventions that could be implemented by government and non-government organisations (NGOs) to make incontinence management easier in LMICs. This includes interventions that are product-based, infrastructural, service-based, health practitioner-targeted, or community-level awareness initiatives.

## Limitations

This study was not able to include any older people with incontinence given that potentially eligible individuals were not willing to participate or acknowledge their condition. This is similar to findings of other studies which indicate that accepting incontinence in older age can be difficult and is an important first step in effective management [33, 34]. This study also purposely selected people with severe incontinence and excluded people with incontinence that could be rectified though medical or surgical intervention. However, there may be a great deal to learn from people with milder incontinence and this may be a much larger proportion of the population. While this study provides insight into the barriers and management of

**Table 4. Description of 5 different domains of interventions that could be implemented by government or non-government actors to improve incontinence management in LMICs.**

| Type of Intervention | Description |
|---|---|
| Product-based Interventions | This may include the local production of acceptable and reusable incontinence products; subsidies or vouchers for purchasing incontinence products (including soap); or the distribution of incontinence management kits which may include products such as scented soap, laundry detergent, cleaning products, plastic bed sheets, and dark-coloured clothing (so that incontinence stains are not easily visible). The sustainability of these types of product-based initiatives must be considered from the outset. |
| Infrastructural Interventions | Interventions in this category relate to how the home environment can be modified to facilitate easier incontinence management and may include the promotion of accessible and easy to clean bathing facilities, prioritisation of households with a PWI when rolling out water supply systems (particularly subsidies to support on-premises connections), and the development of simple lifting aids to support caregivers with the movement of PWI. |
| Service-based Interventions | This may include a range of health, social and economic support services targeted to either the PWI or the caregiver. The design of such initiatives needs to start by developing more robust processes for identifying people with incontinence in LMICs and their caregivers. This may require the refinement and validation of existing incontinence questionnaires so that these tools are suited to a broad range of LMICs. Programmes should also prioritise connecting PWI and their caregivers to each other and supporting the formation of DPOs related to incontinence. This way experiences can be shared and PWI and caregivers can advise programme implementers on appropriate contextualised incontinence management options. |
| Health practitioner-targeted interventions | Health practitioner-targeted interventions: Ministries of Health should be responsible for identifying local language terms for incontinence and developing training on incontinence for all levels of health care professionals. Health care professionals should also be encouraged to actively ask about incontinence with patients who may be more likely to experience the condition. Basic training on incontinence may need to extend to people working within the private or informal sectors (such as private pharmacies or market vendors) if they are in a position to sell incontinence products or give advice on the condition. |
| Community-level interventions | There is a need to normalise and raise awareness about incontinence and effective local coping strategies. Such initiatives should involve PWI and caregivers to ensure that this is communicated in an acceptable manner. |

incontinence in Sindh Province, it is also possible that some of the more culturally specific practices are not generalizable to other LMIC.

## Conclusion

Consistent with prior research our study highlights that incontinence is a condition with complex health, psychological, social, economic, and cultural implications. These consequences affected not only the daily lives of PWI, but to an almost equal extent, it affected the lives of their caregivers. While some of these challenges related to incontinence management have been acknowledged previously, they appear to be exacerbated in LMICs due a lack of recognition of the condition and the absence of interventions by government or NGOs. Making incontinence management easier in LMIC settings needs to start by strengthening government systems to identify and form support networks for PWI and their caregivers. Appropriate solutions need to be developed in partnership with PWI and caregivers and contextualised not just by country, but potentially at a household level to meet individual needs and preferences. Incontinence management initiatives need to be designed for durability, affordability and sustainability and additional research is needed to document and evaluate feasible initiatives to support PWI and caregivers in LMICs.

## Acknowledgments

We would like to thank our research participants for their time and willingness to speak openly about living with incontinence. This research was made possible through the support of Benedicte Hafskjold, Åshild Skare and Corinna Schüttler-Kvarme at Norwegian Church Aid's (NCA) headquarters and Sumera Izhar, Zohaib Hassan, Shah Khalid from NCA in Pakistan who supported with data collection, logistics and access to the study sites.

## Author Contributions

**Conceptualization:** Sian White.

**Data curation:** Zara Ansari.

**Formal analysis:** Zara Ansari.

**Funding acquisition:** Sian White.

**Investigation:** Zara Ansari.

**Methodology:** Zara Ansari, Sian White.

**Supervision:** Sian White.

**Writing – original draft:** Zara Ansari.

**Writing – review & editing:** Sian White.

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
