## [Decision Letter · Decision Letter 0]

2 Mar 2022

PONE-D-22-04296Managing incontinence in low-and middle income-countries: A qualitative case study from PakistanPLOS ONE

Dear Dr. White,

Thank you for submitting your manuscript to PLOS ONE. After careful consideration, we feel that it has merit but does not fully meet PLOS ONE’s publication criteria as it currently stands. Therefore, we invite you to submit a revised version of the manuscript that addresses the points raised during the review process. You'll see both reviewers have some minor, specific comments which it should be straightforward to address, so we look forward to seeing the revised manuscript soon.

We look forward to receiving your revised manuscript.

Kind regards,

Alison Parker

Academic Editor

PLOS ONE

Journal Requirements:

2. We note that Figures 1 to 3 includes an image of a [patient / participant / in the study]. 

As per the PLOS ONE policy (http://journals.plos.org/plosone/s/submission-guidelines#loc-human-subjects-research) on papers that include identifying, or potentially identifying, information, the individual(s) or parent(s)/guardian(s) must be informed of the terms of the PLOS open-access (CC-BY) license and provide specific permission for publication of these details under the terms of this license. Please download the Consent Form for Publication in a PLOS Journal(http://journals.plos.org/plosone/s/file?id=8ce6/plos-consent-form-english.pdf). The signed consent form should not be submitted with the manuscript, but should be securely filed in the individual's case notes. Please amend the methods section and ethics statement of the manuscript to explicitly state that the patient/participant has provided consent for publication: “The individual in this manuscript has given written informed consent (as outlined in PLOS consent form) to publish these case details”. 

Reviewers' comments:

Reviewer's Responses to Questions

**Comments to the Author**

1. Is the manuscript technically sound, and do the data support the conclusions?

Reviewer #1: Yes

Reviewer #2: Yes

2. Has the statistical analysis been performed appropriately and rigorously? 

Reviewer #1: N/A

Reviewer #2: N/A

3. Have the authors made all data underlying the findings in their manuscript fully available?

Reviewer #1: Yes

Reviewer #2: Yes

4. Is the manuscript presented in an intelligible fashion and written in standard English?

Reviewer #1: Yes

Reviewer #2: Yes

5. Review Comments to the Author

Reviewer #1: A valuable study and contribution to the emerging field of incontinence in LMICs and an enjoyable read. The Photovoice aspect was very powerful and of great interest to me. I have a few minor recommendations.

In the introduction to the paper, I would add a brief line about the fact that incontinence can affect anyone at any life stage. A good reference would be the Frontiers of Sanitation on incontinence produced by the Sanitation Learning Hub. This would give context to that fact that whilst this is a study on older people, anyone can be affected.

The section about the study site appears to read as though it is about Pakistan as a whole rather than Sindh province. It would be good to see a little more detail about the locality in general and its WASH context to paint a better picture of the conditions the people with incontinence are living in.

The methodology could be strengthened with a little more detail. Were there any challenges faced in the data collection process? Incontinence is a taboo issue and there can be ambiguity over definitions. How was incontinence defined in Urdu, and were there any issues in getting people to talk about it? A few words about the 6 step analysis method would help the reader to easily follow through. Was there any particular apparent reason why the older people without impairment refused to participate in the study?

On page 8, line one 163, it may be clearer to replace the word ‘predicted’ with ‘planned around’

The section on not being clean for God is really powerful. Interesting to see the religious diversity in this context. How were the minority Hindu families accessed and were there any more details on why they felt they needed to keep clean in the context of their religion? Are there geographical factors which made it easier to access them?

The section on use of local products for incontinence management jumps straight into findings from rural areas. Were there any patterns or distinct differences between urban and rural which can be outlined at the start as an overall pattern?

The five domains of interventions may read easier in the form of a simple and small table with one column for the type of intervention and a second column for description. This is very important information that should stand out. The recommendations would also be strengthened further by adding a little clarity on which actors should be responsible for implementing these.

Reviewer #2: This is a well-written and concise account of a very interesting study. I saw the results presented at a conference several years ago and have been looking forward to its publication!

I have no overarching comments, I think it's great. I have a few small comments on the manuscript, mostly grammatical, but of relevance to the authors a couple of places where I think the citation might be incorrect/a couple of other citations could be added to assist the reader to learn more.

6. PLOS authors have the option to publish the peer review history of their article (what does this mean?). If published, this will include your full peer review and any attached files.

Reviewer #1: **Yes: **Amita Bhakta

Reviewer #2: **Yes: **Dani Barrington

---

## [Author Response · Author response to Decision Letter 0]

13 Jun 2022

Responses to Reviewer #1: 

4. A valuable study and contribution to the emerging field of incontinence in LMICs and an enjoyable read. The Photovoice aspect was very powerful and of great interest to me. I have a few minor recommendations. In the introduction to the paper, I would add a brief line about the fact that incontinence can affect anyone at any life stage. A good reference would be the Frontiers of Sanitation on incontinence produced by the Sanitation Learning Hub. This would give context to that fact that whilst this is a study on older people, anyone can be affected.

Response: Thank you for your kind comments. In the first paragraph we have added that incontinence can affect people across the life cycle. We have also included the reference you suggest. 

5. The section about the study site appears to read as though it is about Pakistan as a whole rather than Sindh province. It would be good to see a little more detail about the locality in general and its WASH context to paint a better picture of the conditions the people with incontinence are living in.

Response: We have edited this section to include more information on the local conditions in Sindh province. 

6. The methodology could be strengthened with a little more detail. Were there any challenges faced in the data collection process? Incontinence is a taboo issue and there can be ambiguity over definitions. How was incontinence defined in Urdu, and were there any issues in getting people to talk about it? A few words about the 6 step analysis method would help the reader to easily follow through. 

Response: We have described how there is no word in Urdu for incontinence and how we went about developing a definition that was used in the study. We have added extra detail to the analysis process. 

7. Was there any particular apparent reason why the older people without impairment refused to participate in the study?

Response: The older people that were approached stated that they did not have incontinence, even though their caregivers confirmed it. We have not added anything to the text here as this was already stated. 

8. On page 8, line one 163, it may be clearer to replace the word ‘predicted’ with ‘planned around’

Response: This has been changed in the text.

9. The section on not being clean for God is really powerful. Interesting to see the religious diversity in this context. How were the minority Hindu families accessed and were there any more details on why they felt they needed to keep clean in the context of their religion? Are there geographical factors which made it easier to access them?

Response: There are certain districts in Sindh that have a higher number of Hindus living in them and those districts were part of the study. We did not purposively sample on the basis of religion so we did not know they were Hindu when we approached them to be part of the study. This detail came out through the interviews. As described in the text through their attitudes towards incontinence because of their religion seemed consistent with Muslim participants. 

10. The section on use of local products for incontinence management jumps straight into findings from rural areas. Were there any patterns or distinct differences between urban and rural which can be outlined at the start as an overall pattern?

Response: Urban participants did not bring up using any unique alternatives (such as the use of straw or sand) for incontinence management. They relied more on water and soap. This has been clarified in the text of the manuscript. 

11. The five domains of interventions may read easier in the form of a simple and small table with one column for the type of intervention and a second column for description. This is very important information that should stand out. The recommendations would also be strengthened further by adding a little clarity on which actors should be responsible for implementing these.

Response: We have taken the reviewers advice and transformed this into a table (Table 4)

Responses to Reviewer #2: 

12. This is a well-written and concise account of a very interesting study. I saw the results presented at a conference several years ago and have been looking forward to its publication! I have no overarching comments, I think it's great. I have a few small comments on the manuscript, mostly grammatical, but of relevance to the authors a couple of places where I think the citation might be incorrect/a couple of other citations could be added to assist the reader to learn more.

Response: These comments have been incorporated throughout the paper.

---

## [Decision Letter · Decision Letter 1]

5 Jul 2022

Managing incontinence in low-and middle income-countries: A qualitative case study from Pakistan

PONE-D-22-04296R1

Dear Dr. White,

We’re pleased to inform you that your manuscript has been judged scientifically suitable for publication and will be formally accepted for publication once it meets all outstanding technical requirements.

Kind regards,

Alison Parker

Academic Editor

PLOS ONE

Additional Editor Comments (optional):

Reviewers' comments:

Reviewer's Responses to Questions

**Comments to the Author**

1. If the authors have adequately addressed your comments raised in a previous round of review and you feel that this manuscript is now acceptable for publication, you may indicate that here to bypass the “Comments to the Author” section, enter your conflict of interest statement in the “Confidential to Editor” section, and submit your "Accept" recommendation.

Reviewer #2: All comments have been addressed

2. Is the manuscript technically sound, and do the data support the conclusions?

Reviewer #2: Yes

3. Has the statistical analysis been performed appropriately and rigorously? 

Reviewer #2: N/A

4. Have the authors made all data underlying the findings in their manuscript fully available?

Reviewer #2: Yes

5. Is the manuscript presented in an intelligible fashion and written in standard English?

Reviewer #2: Yes

6. Review Comments to the Author

Reviewer #2: (No Response)

7. PLOS authors have the option to publish the peer review history of their article (what does this mean?). If published, this will include your full peer review and any attached files.

Reviewer #2: **Yes: **Dani Barrington

---

## [Editor Report · Acceptance letter]

7 Jul 2022

PONE-D-22-04296R1 

Managing incontinence in low-and middle income-countries: A qualitative case study from Pakistan 

Dear Dr. White:

I'm pleased to inform you that your manuscript has been deemed suitable for publication in PLOS ONE. Congratulations! Your manuscript is now with our production department. 

Kind regards, 

on behalf of

Dr. Alison Parker 

Academic Editor

PLOS ONE